# Sleep drive reconfigures wake-promoting clock circuitry to regulate adaptive behavior

**Markus K. Klose**, **Paul J. Shaw** *

Department of Neuroscience, Washington University School of Medicine, St. Louis, Missouri, United States of America

* shawp@wustl.edu

**Data Availability Statement:** All relevant data are within the paper and its Supporting Information files

**Funding:** This work was funded by the National Institute of Neurological Disorders and Stroke

## Abstract

Circadian rhythms help animals synchronize motivated behaviors to match environmental demands. Recent evidence indicates that clock neurons influence the timing of behavior by differentially altering the activity of a distributed network of downstream neurons. Downstream circuits can be remodeled by Hebbian plasticity, synaptic scaling, and, under some circumstances, activity-dependent addition of cell surface receptors; the role of this receptor respecification phenomena is not well studied. We demonstrate that high sleep pressure quickly reprograms the wake-promoting large ventrolateral clock neurons to express the pigment dispersing factor receptor (PDFR). The addition of this signaling input into the circuit is associated with increased waking and early mating success. The respecification of PDFR in both young and adult large ventrolateral neurons requires 2 dopamine (DA) receptors and activation of the transcriptional regulator *nejire* (cAMP response element-binding protein [CREBBP]). These data identify receptor respecification as an important mechanism to sculpt circuit function to match sleep levels with demand.

## Introduction

Circadian rhythms help animals synchronize motivated behaviors with salient events such as when food or mates are available or when it is time to forage or sleep [1]. However, the optimal time of day to engage in particular behaviors can vary depending upon seasonal and environmental factors, which can, under some circumstances, place competing behaviors in conflict (e.g., foraging versus risk of predation; sleep versus mating success) [2–4]. The mechanisms regulating the timing of competing behaviors are not well understood.

Central and peripheral clocks can be synchronized by sleep and environmental cues such as light, temperature, food, social, and interactions [5–9]. Interestingly, recent evidence indicates that clock neurons do not act in a hierarchical manner but rather regulate behavior as a distributed network [10–15]. Although the regulation of these networks is complex, the neuropeptide pigment dispersing factor (PDF) and its receptor (PDFR) play a prominent role in synchronizing oscillations in the clock network [13,16]. Indeed, PDF can influence the timing of behavior by differentially staggering the timing of activity peaks in diverse neuronal groups [16,17]. Thus, the PDFR is well suited for regulating the timing of competing behaviors.

In mammals, exposure to short- and long-day photoperiods, which mimic naturally occurring seasonal changes, results in respecification of transmitters and their receptors [18,19].

NR01-NS051305-01A1 and R01NS057105 to PJS. The funders had no role in study design, data collection and analysis, decision to publish, or preparation of the manuscript.

**Competing interests:** The authors have declared that no competing interests exist.

**Abbreviations:** CREBBP, cAMP response element-binding protein; DA, dopamine; LD, light:dark; LNd, dorsal lateral neuron; l-LNv, large ventral lateral neuron; Oa, octopamine; PDF, pigment dispersing factor; PDFR, pigment dispersing factor receptor; RNAi, RNA interference; ROI, region of interest; s-LNv, small ventral lateral neuron; SNAP, sleep-nullifying apparatus; *SuUR, Suppressor of Under-Replication*; WMZ, wake maintenance zone; ZT, Zeitgeber time.

Receptor respecification is a form of plasticity that, like Hebbian and homeostatic plasticity, may be employed to alter circuit function in response to changing environmental demands [19]. Importantly, sleep circuitry is plastic and can change through developmental and in response to environmental factors (e.g., starvation, predation risk, and mating status) [3,20–27]. Surprisingly, it remains unknown whether receptor respecification plays a role in modulating sleep plasticity.

In the *Drosophila* brain, there are approximately 150 clock neurons that are divided into 6 groups [28]. PDF is expressed in both the large and small ventral lateral neurons (l-LNvs and s-LNvs); In contrast to other clock neurons, the l-LNvs are not believed to express PDFR [13,29–31]. Given the role that PDF plays in coordinating the timing of diverse neuronal groups important for adaptive behavior, we hypothesized that under some circumstances, the PDFR could be respecified to help regulate the timing of competing behaviors. Indeed, we find that the PDFR is respecified in the l-LNvs for the first approximately 48 h after eclosion, when sleep drive is highest. Gain and loss of function experiments reveal that in young flies, PDFR expression is associated with increased waking and early mating success. Importantly, the PDFR can be reestablished in adult l-LNvs through prolonged sleep disruption. The most common forms of respecification alter the polarity of the synapse to alter the function of the circuit [19]. In contrast, our data suggest an additional type of respecification in which an input pathway into a circuit can be turned on and off, without changing the sign of the synapse (excitatory/inhibitory). These data identify receptor respecification as an important mechanism to sculpt circuit function to match sleep need with environmental demands.

## Results

### PDFR is expressed in l-LNvs in young flies

Sleep is highest in young animals during a critical period of brain development when neuronal plasticity is high [26,32]. As previously described, flies were collected and sexed using $CO_2$ anesthesia on the day they eclosed, placed into tubes, and sleep was quantified during their first full day of adult life. Sleep is highest during the first 48 h after eclosion (day 0, day1) and then reaches stable mature adult levels by approximately day 3 (Fig 1A and 1B). The increased sleep observed during these approximately 48 h is important for the development of circuits that maintain adaptive behavior into adulthood [33,34]. How neurons in sleep circuitry change during this period has not been explored. The l-LNvs promote waking behavior through both dopamine (DA) and octopamine (Oa) signaling (19,22–24); thus, we hypothesized that one or both of these pathways might be down-regulated during this early developmental period of high sleep. To test this hypothesis, we used live brain imaging in l-LNvs expressing the reporter Epac1 camps to define cAMP response properties [30,31,35,36]. Contrary to our hypothesis, neither DA- or Oa-induced cAMP responses changed as the flies matured (Figs 1C and 1D and S1). Interestingly, we did observe PDF-induced cAMP responses in l-LNvs in the first 48 h of adulthood (Fig 1E and 1F), while they were predominantly absent in mature adult l-LNvs, consistent with previous reports [30,31]. To determine if this transient PDF sensitivity is regulated at the receptor level, expression of the PDFR was examined directly using *Pdfr-myc*, a tagged receptor genetic construct under the natural PDF promoter [29]. As anticipated, detection of MYC antibody staining is high on day 0 and not detectable on day 5 of adulthood (Fig 1G), revealing transient expression of the receptor. Finally, we examined an adjacent group of clock neurons, the s-LNvs [37]. Responses to PDF in s-LNvs are present at the beginning of adulthood and then decrease in amplitude over the first approximately 48 h of adulthood. In contrast to the l-LNvs, sensitivity to PDF in the s-LNvs persists into mature adulthood (S1

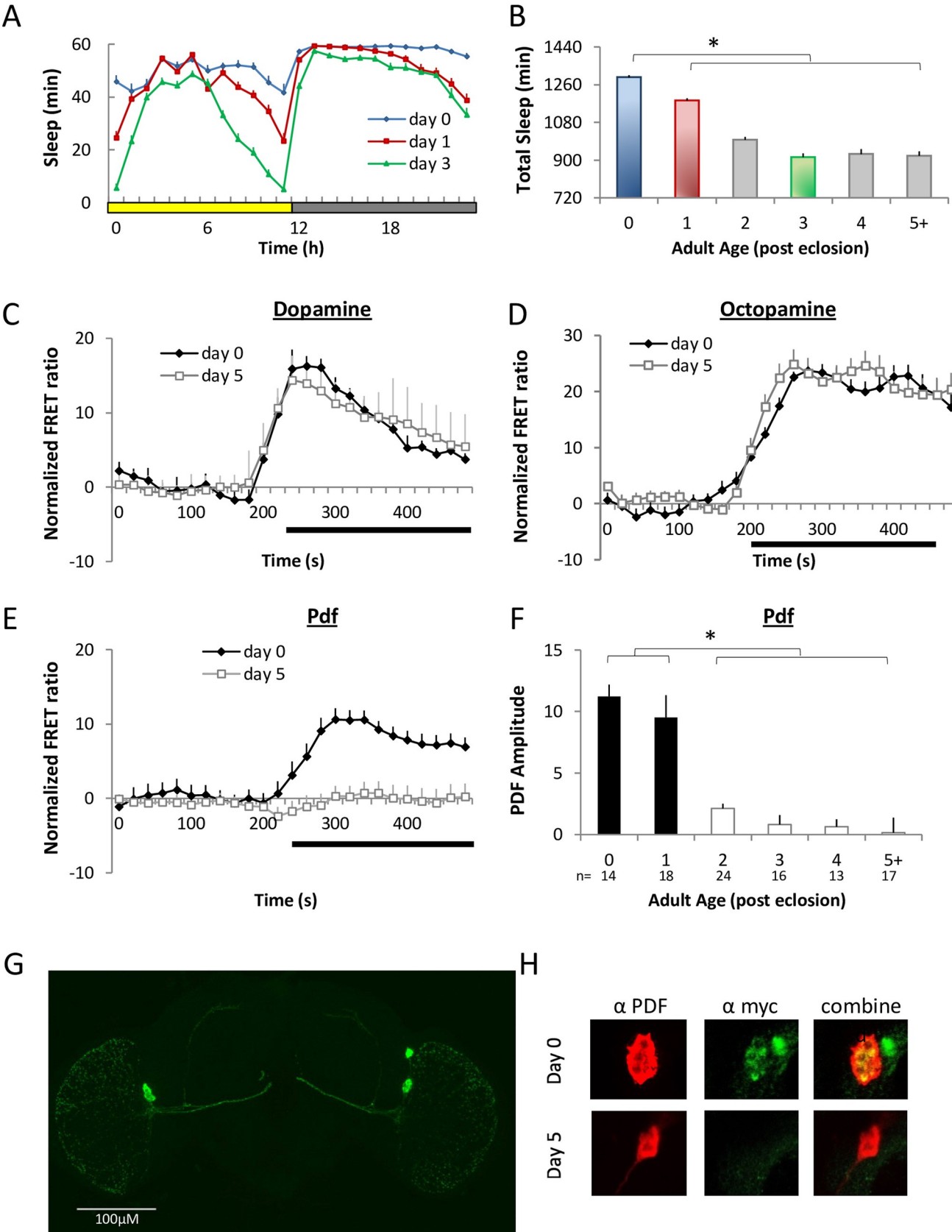

**Fig 1. PDFR is expressed in l-LNvs of young flies. (A, B)** Sleep is elevated in young male flies following eclosion and reaches stable adult values in 3-day-old flies ($n$ = 35–93 flies/age; one-way ANOVA $F_{[5,472]}$ = 81.34, for age, $p$ = 3.7$^{E-63}$). **(C–E)** FRET ratio measurements in *Pdf-GAL4>UAS-Epac1* flies in response to DA (3e-3M), Oa (3e-3M), and PDF (1e-6M) ($n$ = 5–15 ROI. Each ROI represents 2 to 4 l-LNvs). **(F)** The amplitude of l-LNv responses to PDF decreases with age (PDF amplitude); ($n$ = 13–24 ROI/age; one-way ANOVA for age $F_{[5,93]}$ = 19.86, $p$ = 3.3$^{E-13}$). **(G)** GFP expression in LNv neurons (*Pdf-GAL4*; UAS-*gfp*). **(H)** Immunohistochemistry reveals coexpression of PDF (red) and myc (green) in 0-day-old *P[acman] Pdfr-myc70* flies, which is not observed on day 5. $^*p < 0.05$, modified Bonferroni test. Data underlying this figure can be found in S1 Data. DA, dopamine; FRET, Förster Resonance Energy Transfer; GFP, green fluorescent protein; l-LNv, large ventral lateral neuron; Oa, octopamine; PDF, pigment dispersing factor; PDFR, pigment dispersing factor receptor; ROI, region of interest.

Fig). Together, these data indicate that the PDFR is transiently expressed in wake-promoting l-LNvs in young flies to support waking when sleep drive is highest.

## Expression of PDFR in l-LNvs alters behavior in young flies

During the day, sleep is highest during the midday siesta and is reduced in the hours preceding lights out (Fig 1A) [26,33,34]. We have operationally defined the 2-h period before lights out as the wake maintenance zone (WMZ) based upon the observation that sleep rebound is absent or dramatically reduced when flies are released into recovery during this time window [33,38]. The ability to maintain waking in the face of high sleep drive suggests that this window of time is protected for important waking behaviors [39]. With that in mind, we hypothesized that flies lacking the PDFR would sleep more than genetic controls during the WMZ. *Pdfr*-null mutant (*Pdfr$^{5304}$*) flies were outcrossed to *Cs* flies for 5 generations. To avoid handling of flies on the day they eclosed, *Pdfr$^{5304}$* and *Cs* flies were plated on juice plates for 4 h to lay eggs, and then L1 larvae were put into individual glass tubes and monitored. Sleep was assessed in male flies that eclosed between Zeitgeber time (ZT1) and ZT4. As seen in Fig 2A and 2C on the day of eclosion, *Pdfr$^{5304}$* null mutants sleep significantly more than their genetic controls during the WMZ. To determine whether the change in sleep was due to expression of the PDFR in the l-LNvs, we expressed wild-type *Pdfr* (*UAS-Pdfr$^{wt}$*) using the *c929-GAL4* driver in a *Pdfr$^{5304}$* mutant background. Since *c929-GAL4* is expressed in other peptidergic neurons [40], we combined *c929-GAL4* with *cry-Gal80*, which targets the *GAL4* inhibitor *GAL80* to all CRY+ neurons including the l-LNvs to suppress the PDFR rescue [41]. As seen in Fig 2B and 2D, sleep remained elevated during the WMZ in *Pdfr$^{5304}$;c929/+;cryGAL80/+* (green) and *Pdfr$^{5304}$;UAS-Pdfr$^{wt}$/+* (purple) parental controls as expected. In contrast, waking was rescued during the WMZ in *Pdfr$^{5304}$;c929/UAS-Pdfr$^{wt}$* flies (red), and this increase in waking was prevented when the expression of *UAS-Pdfr$^{wt}$* was blocked in clock cells (*Pdfr$^{5304}$;c929/ UAS-Pdfr$^{wt}$;cryGAL80/+*,blue). We verified the effectiveness of *cry-GAL80* using a *UAS-GFP* reporter (S2 Fig). To further exclude the possibility that expression of *UAS-Pdfr$^{wt}$* in other peptidergic neurons outside the l-LNvs altered waking, we rescued the expression *UAS-Pdfr$^{wt}$* in a *Pdfr$^{5304}$* mutant background using *Pdf-GAL4*, which targets only LNv neurons. As seen in S2 Fig, sleep was reduced in *Pdfr$^{5304}$;Pdf-GAL4/UAS-Pdfr$^{wt}$* compared to parental controls. Finally, we asked whether the inability of *Pdfr$^{5304}$* mutants to stay awake during the WMZ was due to the absence of *Pdfr* in the *l-LNvs*. As seen in Fig 2E, *Dcr2;929-GAL4/ UAS-Pdfr$^{RNAi}$* flies slept significantly longer during the WMZ than either *Dcr2;c929-GAL4/+* or *UAS-Pdfr$^{RNAi}$/+* parental controls. Together, these data indicate that PDFR in the l-LNv promotes waking in young flies when sleep drive is high.

Although the respecification of the PDFR in l-LNvs supports waking in young flies, it is unclear whether the observed changes impact ecologically relevant behaviors. Inspired by the observation that the male pectoral sandpipers that sleep the least during breeding season sire more offspring [3], we assayed mating success in flies with and without PDFR. As above, we began by evaluating *Pdfr$^{5304}$* mutants and their genetic controls (*Cs*). Following eclosion, male flies were individually paired with a wild-type virgin female fly at ZT4 for 20 h, and the

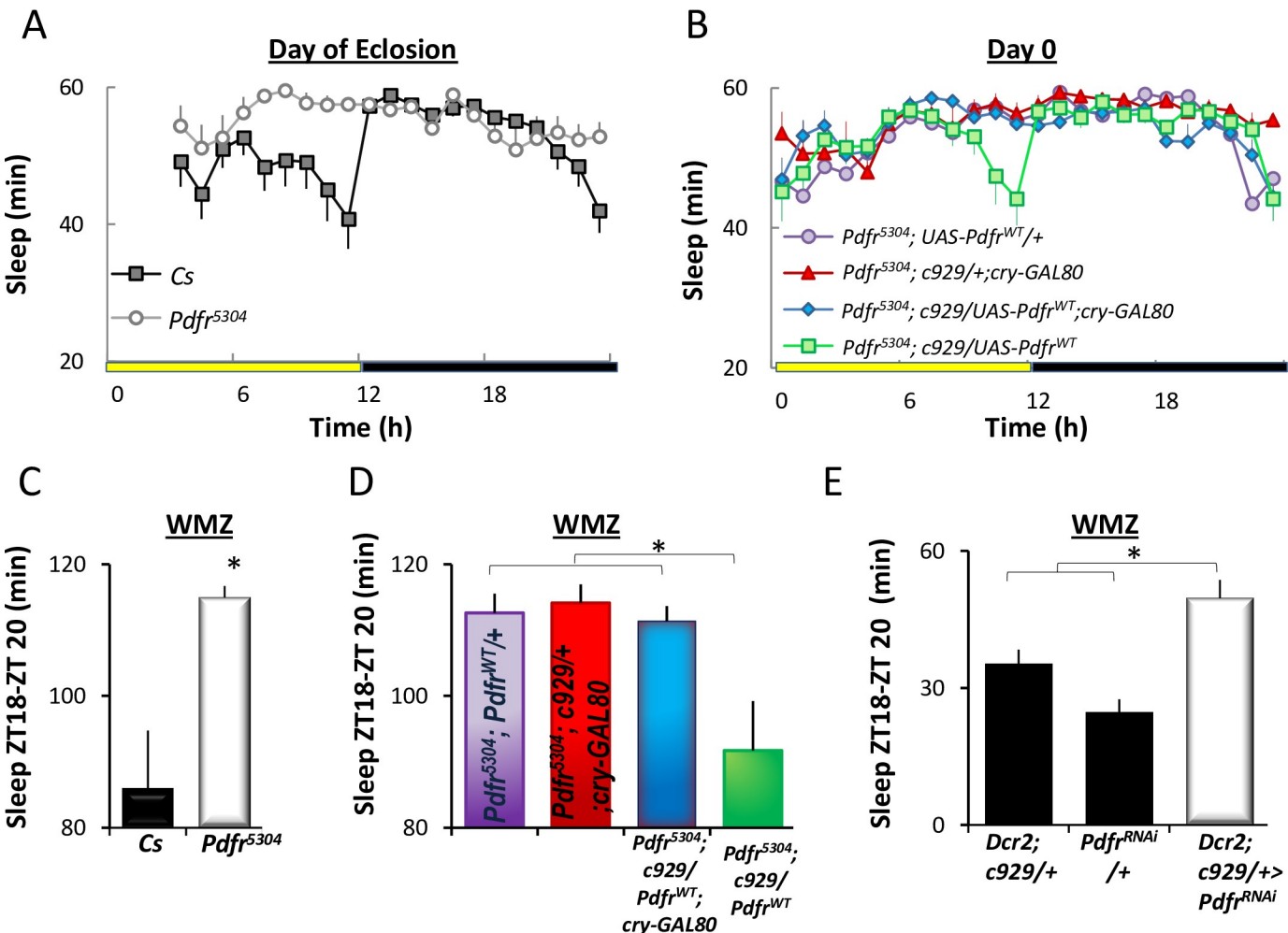

**Fig 2. Expression of PDFR in l-LNvs regulates sleep in young flies.** (A) Sleep traces of *Pdfr5504* mutants and *Cs* controls on day 0. (B) Sleep traces for *Pdfr5504*; *c929-GAL4; UAS-Pdfr* (rescue, green), *Pdfr5504*; *UAS-Pdfr/+*, *Pdfr5504*; *c929-GAL4/+; Cry-Gal80/+*, and *Pdfr5504*; c929-GAL4/UAS-Pdfr; Cry-Gal80 (n = 26–31/ genotype). (C) Quantification of sleep during the WMZ of flies shown in (A). *Cs* flies sleep less during the WMZ than *Pdfr5504* mutants (n = 26/genotype; t test, p < 0.05); (D) Quantification of sleep during the WMZ of flies shown in (B). *Pdfr5504*; *c929-GAL4; UAS-Pdfr* sleep less than parental controls; ANOVA F$_{[3,109]}$ = 6.33, p = 5.4$^{E-4}$; n = 22–31. (E) Sleep is increased in *Dcr2; c929-GAL4/UAS-Pdfr*$^{RNAi}$ flies on day 0 compared to *Dcr2; c929-GAL4/+* and *UAS-Pdfr/+* parental controls (ANOVA; F$_{[2,279]}$ = 12.00, p = 1.04$^{E-5}$; n = 26–28). Data underlying this figure can be found in S2 Data. l-LNv, large ventral lateral neuron; PDFR, pigment dispersing factor receptor; WMZ, wake maintenance zone.

pairings that produce offspring were tabulated. As seen in Fig 3A, approximately 80% of pairings with Cs males resulted in offspring, while only 25% of pairings with *Pdfr5504* mutants were successful on day 0. Moreover, mating success was also reduced when Pdfr was knocked down in *c929-GAL4* expressing cells (Fig 3B). Importantly, the deficit in mating success observed in *Pdfr5504* mutants on day 0 was rescued by expressing wild-type PDFR using c929-GAL4 (Fig 3C). Previous studies have shown that the expression of PDFR in the dorsal lateral (LNd) neurons, a different set of clock neurons, promotes mating behavior in mature males [42]. However, no changes in mating success were observed in 2-day-old *Pdfr5504* mutants or in *Pdfr5504*; *c929/UAS-Pdfr*$^{wt}$ rescue flies compared to genetic controls (Fig 3A–3C).

To further determine whether expression of PDFR in the l-LNvs was important for mating success, we utilized a competition assay in which we rescued PDFR in a *Pdfr5504* mutant background. In this assay, one red-eyed male and one white-eyed male were combined with a

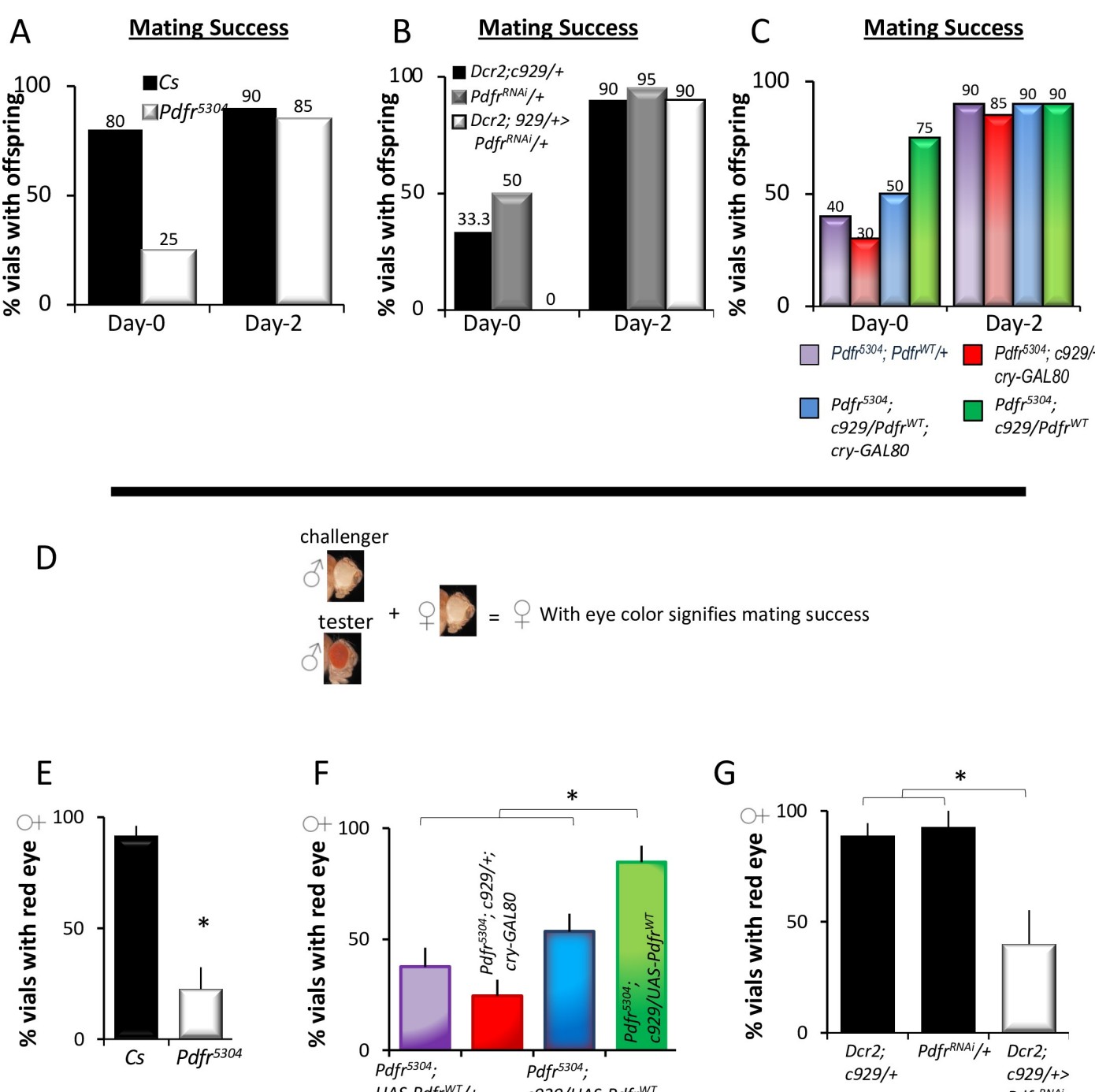

**Fig 3. Role for PDFR in l-LNvs in mating success.** (**A**) *Cs* flies produce more offspring than *Pdfr^5504* mutants (*n* = 30/genotype); χ² = 12.13, *p* = 0.0004. (**B**) *Dcr2; c929*-GAL4/UAS-*Pdfr^RNAi* flies produce fewer vials with offspring compared to *Dcr2; c929*-GAL4/+ and +/UAS-*Pdfr^RNAi* parental controls; (*n* = 30/genotype); χ² = 21.3, *p* = 0.00002 (**C**) *Pdfr^5504*; *c92*-GAL4; UAS-*Pdfr^WT* (rescue, green) flies produce more offspring than *Pdfr^5504*; UAS-*Pdfr^WT*/+ (purple), *Pdfr^5504*; *c929*-GAL4/+; *CryGal80*/+ (red), and *Pdfr^5504*; *c929*-GAL4/UAS-*Pdfr*; *Cry*-Gal80 (blue) parental controls (*n* = 30/genotype) χ² = 8.95, *p* = 0.029. (**D**) Mating competition assay scheme on day 1. (**E**) Cs males outcompeted white-eyed challenger flies compared to *Pdfr^5504* mutants (*t* test, *p* < 0.001, *n* = 3 sets of 20 flies/genotype). (**F**) *Pdfr^5504*; *c929*-GAL4/+; UAS-*Pdfr*/+ males outcompeted white-eyed challengers compared to *Pdfr^5504*; UAS-*Pdfr*, *Pdfr^5504*/+; *c929*-GAL4/+; *Cry*-Gal80/+, or *Pdfr^5504*; *c929*-GAL4/UAS-*Pdfr*; *Cry*-Gal80/+ controls (ANOVA F[3,13] = 15.01, *p* = 4.9⁻⁴; *n* = 3 sets of 20 flies/line). (**G**) *Dcr2; c929*-GAL4/UAS-*Pdfr* RNAi flies displayed reduced mating success compared to *Dcr2; c929*-GAL4/+ and +/UAS-*Pdfr* RNAi control flies (ANOVA, F[2,8] = 8.10, *p* = 0.019, *n* = 43–55). *p* < 0.05, modified Bonferroni test. Data underlying this figure can be found in S3 Data. l-LNv, large ventral lateral neuron; PDFR, pigment dispersing factor receptor; RNAi, RNA interference.

white-eyed female for 2 h at the beginning of the WMZ at ZT10 on day 1. Successful mating of the red-eyed male was determined by female progeny with eye color (Fig 3D). Consistent with the data presented above, *Pdfr⁵³⁰⁴* males sired fewer offspring than the Cs controls (Fig 3E). Despite the fact that *white⁻* flies show impaired courtship [43,44], white-eyed males sired more offspring than red-eyed *Pdfr⁵³⁰⁴;c929/+*, *Pdfr⁵³⁰⁴; UAS-Pdfrʷᵗ*, and *Pdfr⁵³⁰⁴;c929/UAS-Pdfrʷᵗ; cryGAL80/+* controls (Fig 3F). In contrast, male flies expressing the Pdfr in l-LNvs (*Pdfr⁵³⁰⁴; c929/UAS-Pdfrʷᵗ*) sired more red-eyed progeny on day 1 (Fig 3F). To determine whether the deficit in mating success in *Pdfr⁵³⁰⁴* mutants was due to loss of PDFR in the l-LNvs, we evaluated *Dcr2;929-GAL4/UAS-Pdfrᴿᴺᴬⁱ* flies. As seen in Fig 3G, *Dcr2; c929>; UAS-Pdfrᴿᴺᴬⁱ* lines reduced mating success compared to *Dcr2; c929/+* and UAS-*Pdfrᴿᴺᴬⁱ/+* parental controls. Therefore, the expression of the PDF receptor in the l-LNvs is associated with successful mating in early adulthood when sleep pressure is high.

## Respecification of PDFR in l-LNvs modulates adult behavior

Given that the expression of the PDFR in the l-LNvs confers advantages to the young fly, we wondered why its expression would then be turned off on days 2 to 3 of adult life. To gain further insight into this question, we evaluated behavior in 5-day-old flies ectopically expressing the PDFR in the l-LNv using a specific split-GAL4 driver (*GRSS000645*, *l-LNv-GAL4*). Daytime sleep was modestly reduced in *l-LNv-GAL4>UAS-Pdfrʷᵗ* flies compared to *l-LNv-GAL4/+* and *UAS-Pdfrʷᵗ/+* parental controls (Fig 4A). As a negative control, we evaluated sleep in adult

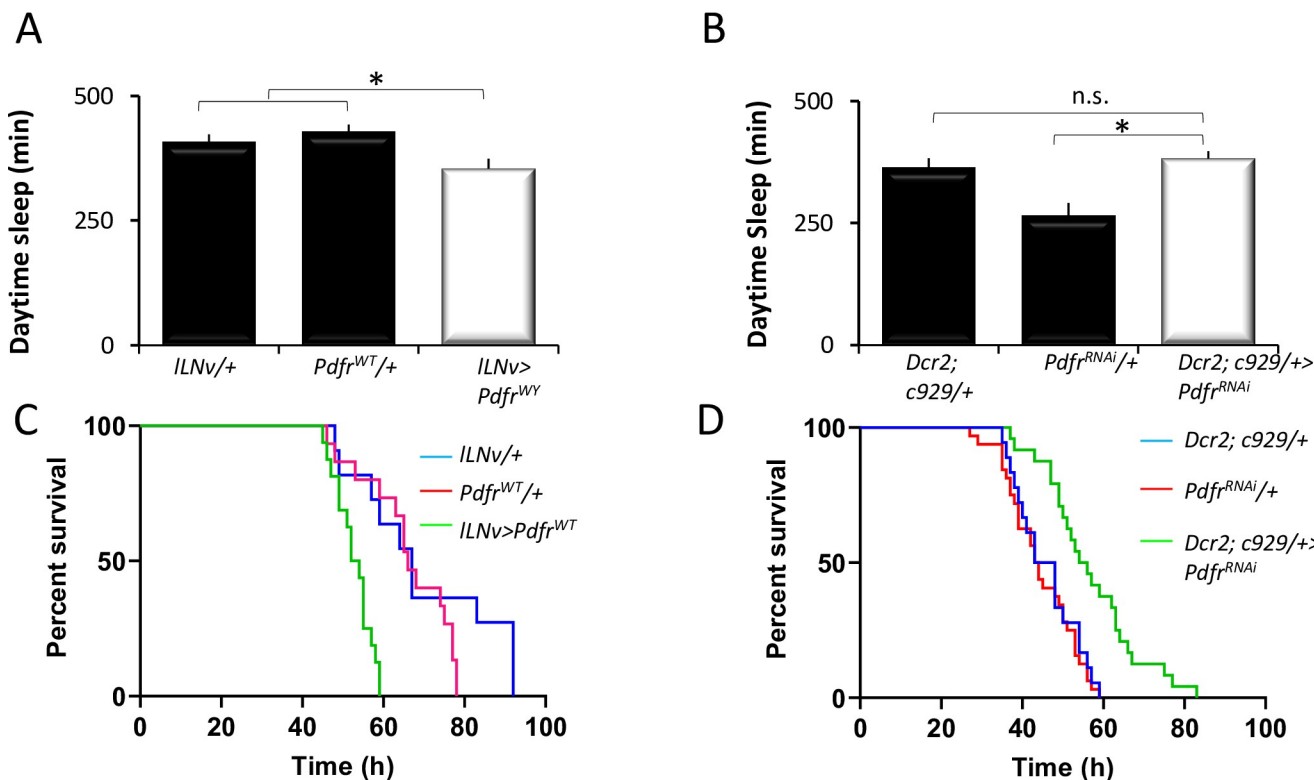

**Fig 4. Behavioral consequences of PDFR expression in l-LNvs. (A)** Daytime sleep in 5-day-old *l-LNv-GAL4>; UAS-Pdfrʷᵀ/+* flies and parental controls (ANOVA[2,92] = 5.7 p = 0.004; n = 30–32/genotype). **(B)** Sleep in *Dcr2; c929*-GAL4/UAS-*Pdfr* flies and parental controls (ANOVA[2,150] = 17.24 p = 1.86ᴱ⁻⁰⁷; n = 40–60/genotype; *p < 0.05, modified Bonferroni test). **(C)** Kaplan–Meier analysis reveals % survival during starvation in *l-LNv-GAL4>; UAS-Pdfrʷᵀ* flies and parental controls (n = 3 replicates of 10–16/genotype, χ² = 19.55, df = 2, p-value < 0.0001). **(D)** Kaplan–Meier analysis reveals % survival during starvation in *Dcr2; c929*-GAL4/UAS-*Pdfrᴿᴺᴬⁱ* flies and parental controls (n = 3 replicates of 10–16/genotype, χ² = 23.35, df = 2, p-value < 0.00001). Data underlying this figure can be found in S4 Data. l-LNv, large ventral lateral neuron; n.s., not significant; PDFR, pigment dispersing factor receptor.

flies while expressing *UAS-Pdfr*[RNAi] in the l-LNvs. Not surprisingly, expressing *UAS-Pdfr*[RNAi] in the l-LNvs did not alter sleep in adult flies (Fig 4B). Previous studies have shown that mutations that confer resistance in one environmental setting may increase the vulnerability of individuals in alternate settings [45]. Thus, we hypothesized that increased waking could sufficiently alter energy demands to make adult flies expressing PDFR in the l-LNvs vulnerable to starvation. To test this hypothesis, we starved flies and examined survival. As above, we examine the impact of starvation when the PDFR was overexpressed or knocked down in the l-LNvs. As seen in Fig 4C, survival was shorter in *l-LNv>*UAS-Pdfr[wt] compared to *l-LNv/* + and UAS-Pdfr[wt]/+ parental controls. Astonishingly, *l-LNv-GAL4>UAS-Pdfr*[RNAi] flies showed improved survival compared to both parental controls flies (Fig 4D). Although knocking down *Pdfr* attenuated starvation induced waking, sleep was reduced in both *pdf*[01] mutants and *w*[1118] genetic controls during the first 18 h of starvation (S4 Fig). However, waking activity was significantly lower in *pdf*[01] mutants, and they survived significantly longer compared to *w*[1118] controls (S4 Fig). Together, these data indicate that the ability of *pdf* to influence adaptive behavior during starvation extends beyond modifying sleep time.

The increased survival seen in starved *l-LNv-GAL4>UAS-Pdfr*[RNAi] flies suggested that the genetic program that activates the PDFR in the l-LNvs may be reactivated in mature adults during conditions of high sleep drive. Short periods of starvation (approximately 12 h) increase waking without activating sleep drive presumably to maintain cognition during foraging [45,46]. However, longer periods of starvation (approximately 20 h) are able to activate homeostatic mechanisms [22]. Thus, we hypothesized that starvation would lead to the respecification of the PDFR in the l-LNvs. As seen in Fig 5A, PDF responses in the l-LNvs of mature adult flies are restored following starvation when compared to age-matched, nonstarved siblings. To determine how much time was required for starvation to respecify the PDFR in the l-LNvs, we evaluated the time course of PDFR respecification in the l-LNvs. Interestingly, starvation-induced restoration of PDF sensitivity in l-LNvs requires a similar duration as reported for the activation of homeostatic drive (Fig 5B). These data suggest that the respecification of the PDFR in the l-LNvs is to help flies maintain wakefulness during starvation. With that in mind, we hypothesized that blocking the expression of the PDFR in the l-LNvs would result in more sleep during starvation. Indeed, *Dcr2; c929>; UAS-Pdfr*[RNAi] flies slept more than parental controls between 21 h of starvation, when homeostatic drive begins, and 32 h of starvation prior to the point when flies begin dying (S5 Fig). In summary, starvation reduces sleep resulting in a buildup of sleep pressure, which may mimic the conditions present in early adulthood that lead to PDFR respecification in l-LNvs.

Starvation is an indirect method to increase sleep pressure. With that in mind, we asked whether sleep deprivation would also result in the respecification of the PDFR in the l-LNvs in mature, adult flies. As seen in Fig 5D, l-LNvs respond physiologically to PDF following sleep deprivation in 5-day-old flies. Although total sleep deprivation is the most common method for increasing sleep drive in the laboratory, it seems unlikely that circumstances in the natural environment would keep an animal awake continuously for 12 h or more. In contrast, sleep consolidation is more easily disrupted and, perhaps, more likely to be impacted by a variety of environmental conditions [21,23]. Thus, we hypothesized that interrupting sleep consolidation would be sufficient to respecify the PDFR in the l-LNvs. A variety of manipulations that increase sleep drive (e.g., memory consolidation and activating the dorsal fan-shaped body) increase average daytime sleep bout duration to >22 min/bout [47,48]. Thus, we disrupted sleep consolidation by presenting a mechanical stimulus to the flies for 1 min every 15 min for 48 h. As seen in Fig 5C, this protocol modestly disrupted sleep and did not result in a compensatory sleep rebound. To determine if the lack of a sleep rebound was due to the respecification of PDFR, we examined the l-LNvs physiologically and histologically. As seen in Fig 5D, PDF

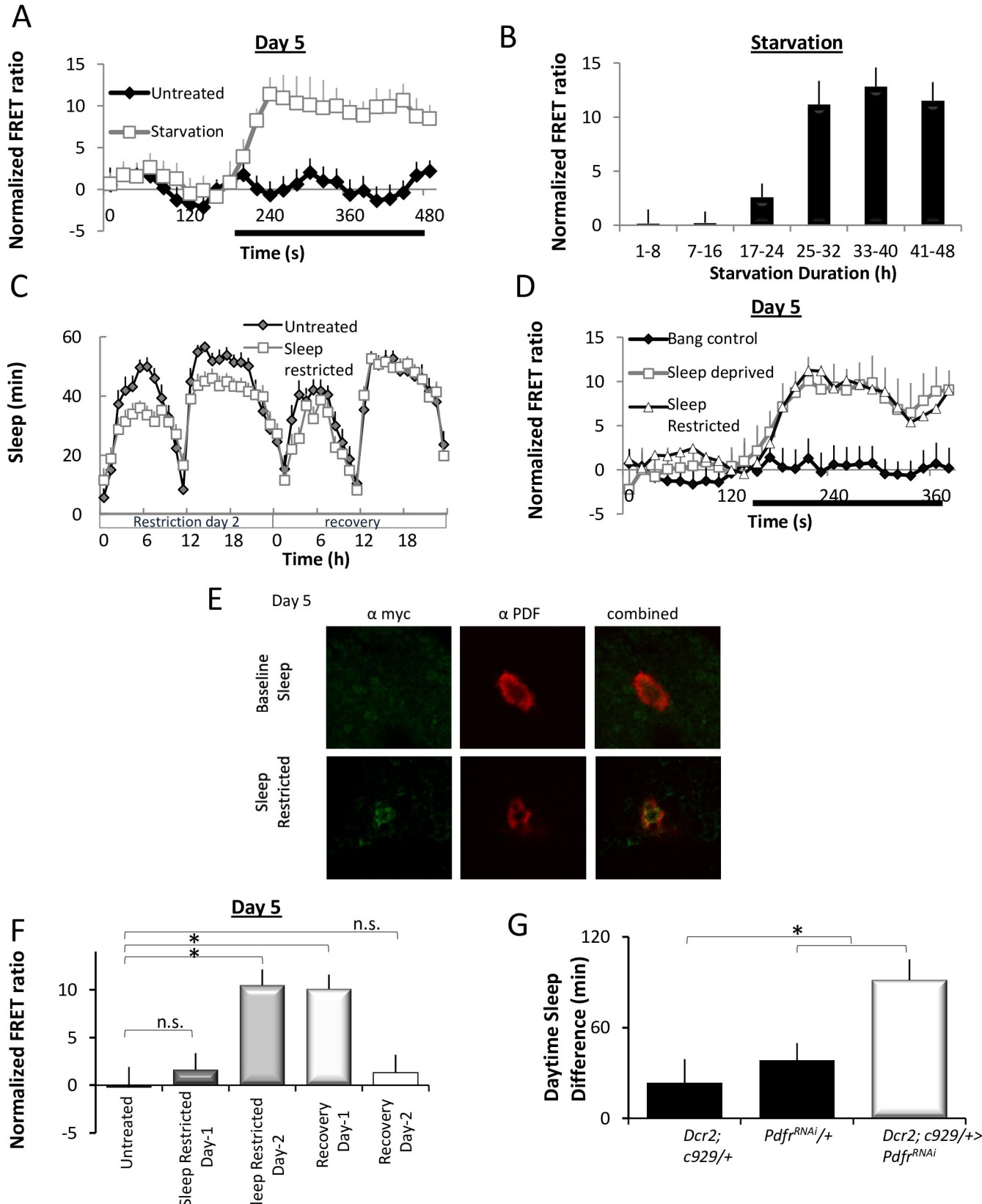

**Fig 5. Sleep pressure induces PDFR expression in mature l-LNvs. (A)** Normalized FRET ratio during PDF application in l-LNvs from starved ($n = 10$) and fed ($n = 16$) *Pdf-GAL4>UAS-Epac1* flies. **(B)**The amplitude of l-LNv responses to PDF is observed in 5-day-old *Pdf-GAL4>UAS-Epac1* flies following 21–24 h of starvation. Data are shown for 8 h bins (ANOVA $F_{[7,71]} = 9.08$, $p = 9.8^{E-8}$; *ROI* = 7–20). **(C)** Sleep in *Cs* flies on sleep restriction day 2 and recovery. **(D)** Normalized FRET ratio during PDF application in l-LNvs recorded from *Pdf-GAL4>UAS-Epac1* flies during sleep restriction sleep deprivation and bang controls ($n = 12$–25 ROI/genotype). **(E)** Immunohistochemistry of PDF (red) and myc (green) in 5-day-old sleep-restricted *P[acman] Pdf-myc70* flies. **(F)** The amplitude of l-LNv responses to PDF in l-LNvs during baseline, sleep restriction, and recovery (ANOVA $F_{[4,81]} = 8.00$, $p = 1.94^{E-4}$, $n = 9$–24 ROI/ condition). **(G)** Sleep rebound in *Dcr2; c929-GAL4/UAS-Pdfr$^{RNAi}$* flies and parental controls (ANOVA $F_{[2,147]} = 6.37$, $p = 2.22^{E-3}$; $n = 43$–55). *$p < 0.05$, modified Bonferroni test. Data underlying this figure can be found in S5 Data. FRET, Förster Resonance Energy Transfer; l-LNv, large ventral lateral neuron; n.s., not significant; PDF, pigment dispersing factor; PDFR, pigment dispersing factor receptor; ROI, region of interest.

responses in the l-LNvs of mature adult flies are restored following 48 h of sleep restriction. To determine if the mechanical stimulus alone would respecify the PDFR in the l-LNvs, siblings were exposed the same amount of stimulation (approximately 190 min) as sleep-restricted siblings but during the biological day when sleep debt does not accrue [26,49]. As expected, mechanical stimulation in the absence of sleep restriction did not respecify the PDFR in the l-LNvs (Fig 5D). To confirm the physiological data, PDFR was examined directly using *Pdfr-myc* [29]. MYC antibody staining in the l-LNvs is clearly visible in mature adult flies following sleep restriction but is not observed in nondisturbed age-matched controls (Fig 5E). Next, we asked how much sleep restriction was required for the respecification of the PDFR. As seen in Fig 5F, PDF sensitivity becomes apparent after 24 to 48 h of sleep restriction, is sustained during the first day of recovery, and then dissipates. Finally, we asked whether knocking down the PDFR in the l-LNvs would modulate sleep homeostasis following sleep disruption. As seen in Fig 5G, *Dcr2;c929/+>UAS-Pdfr$^{RNAi}$* flies slept significantly more following sleep restriction than *Dcr2;c929/+ and UAS-Pdfr$^{RNAi}$/+* parental controls. These data indicate that PDFR can be respecified to mitigate against the effects of sleep pressure in the context of sleep disruption.

## Nejire modulates PDFR in both young and mature l-LNvs

The PDFR is transiently expressed in the l-LNvs of young flies and can be respecified again in mature adults in response to certain environmental perturbations. Thus, we asked whether these seemingly different conditions invoke the same mechanisms to activate the expression of PDFR in the l-LNvs. To begin, we conducted an RNA interference (RNAi) screen of transcription factors that are known to be expressed in LNvs [50]. We crossed UAS-RNAi lines with *pdf-GAL4;UAS-Epac* and monitored PDF sensitivity in both l-LNvs and s-LNvs in young flies on day 0. As mentioned above, s-LNvs display persistent expression of the PDFR in both young and mature adults. Thus, we hypothesized that by monitoring both cell types, we could distinguish between regulatory elements specific to the transient pathway in l-LNvs. We also examined DA responses to discriminate between transcription factors specifically involved in the PDF pathway and those common to other GPCR signaling pathways. As seen in Fig 6A, knocking down *Drosophila cAMP response element-binding protein (CREB)* (*nejire*) or *Suppressor of Under-Replication (SuUR)* ablated PDF sensitivity in l-LNvs on day 0, while other transcription factors left the sensitivity of the l-LNvs to PDFR largely intact. The amplitude of DA responses was not altered by *nejire*, *SuUR*, or any other RNAi lines, revealing that the roles of *nejire* and *SuUR* are specific to the PDF pathway in this context (S6A Fig). PDF sensitivity in the s-LNvs was not ablated by RNAi knockdown of *nejire* (S6 Fig). Interestingly, *nejire* also plays a role in the respecification of the PDFR in the l-LNvs in mature adults following sleep restriction (Fig 6B). As in young flies, the panel of RNAi lines did not alter DA responses in the l-LNvs (S6 Fig). To further evaluate the role of *nejire* in the respecification of the PDFR in mature adults, we expressed wild-type *nejire* (*UAS-nejire$^{WT}$*) or *UAS-nejire$^{RNAi}$* using *Pdf-GA4; UAS-Epac*. We hypothesized that the overexpression of *nejire* would restore PDFR sensitivity to the l-LNvs in well-rested mature adults and that knocking down *nejire* would block

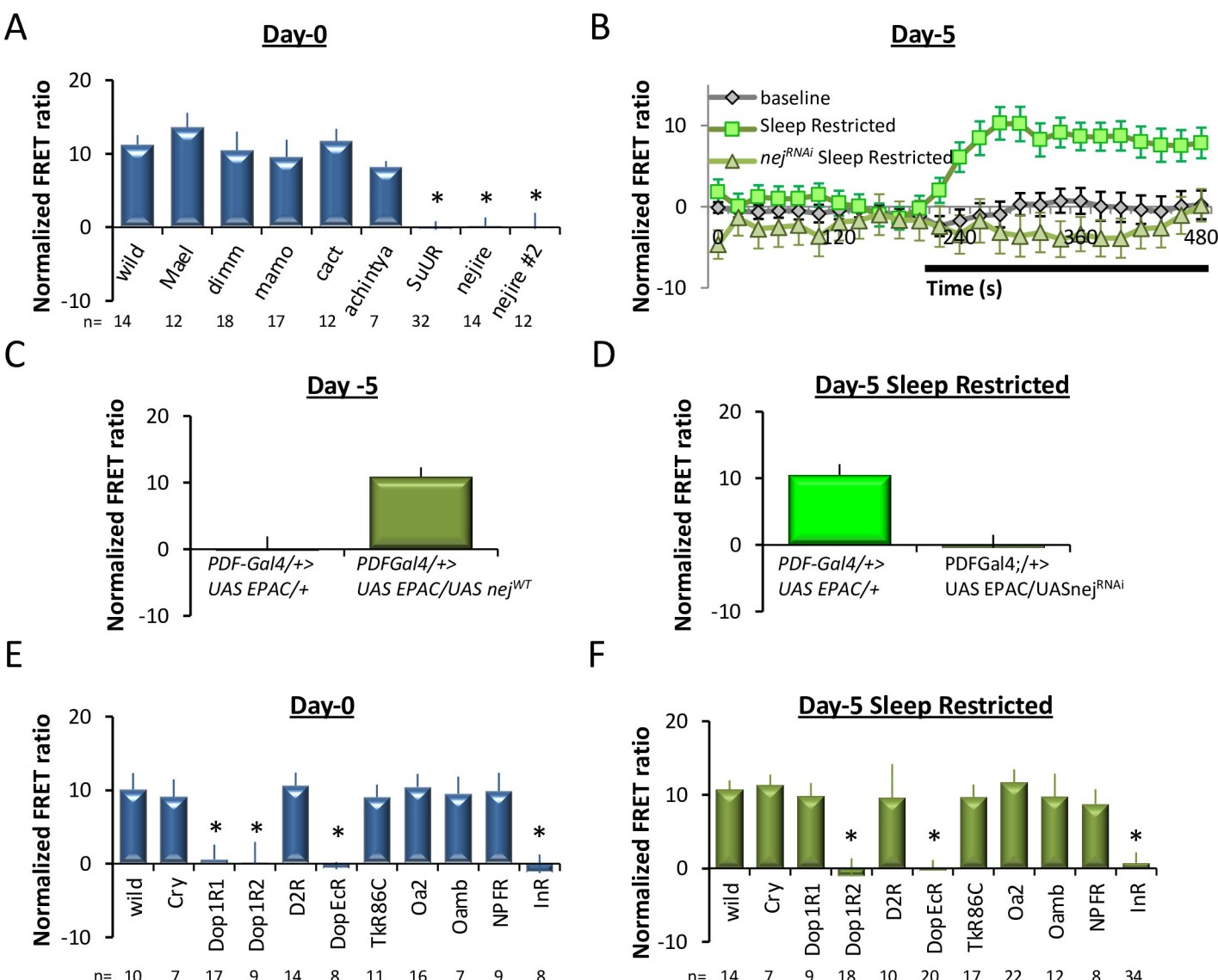

**Fig 6. *nejire* regulates PDFR respecification in l-LNvs of both young and mature flies. (A)** The amplitude of l-LNv responses to PDF on day 0 in *Pdf-GAL4>UAS-Epac1* flies crossed to *UAS-RNAi* lines of the depicted transcription factors (ANOVA; $F_{[8,127]}$ = 11.1, $p$ = 1.26$^{E-11}$, *$p$ < 0.05, modified Bonferroni test, $n$ are listed below the x-axis). **(B)** PDF amplitude in control and 5-day-old sleep restricted *Pdf-GAL4>UAS-Epac1* flies compared to *Pdf-GAL4>UAS-Epac1* flies expressing *UAS-nej$^{RNAi}$* ($n$ = 6–19 neurons/genotype). **(C)** In the absence of sleep loss, the l-LNvs of *Pdf-GAL4>UAS-Epac1/UAS-nej$^{WT}$* respond to PDF while age-matched *Pdf-GAL4>UAS-Epac1* do not (*t* test, DF 34–1, t = 22.6, $p$ = 3.74$^{e-5}$, $n$ = 9–26 neurons/genotype). **(D)** Data quantified from (B). (*t* test, DF 40–1, t = 17.8, $p$ = 4.10$^{e-4}$, $n$ = 16–26). **(E)** The amplitude of l-LNv responses to PDF in l-LNvs on day 0 in *Pdf-GAL4>UAS-Epac1* flies crossed to *UAS-RNAi* lines of the depicted cell surface receptors (ANOVA $F_{[10,105]}$ = 5.38, $p$ = 2.8$^{-6}$ $p$ < 0.05, modified Bonferroni test, $n$ are listed below the x-axis). **(F)** The amplitude of l-LNv responses to PDF in l-LNvs on day 5 in sleep-restricted *Pdf-GAL4>UAS-Epac1* flies crossed to *UAS-RNAi* lines of the depicted cell surface receptors (ANOVA $F_{[10,179]}$ = 7.04, $p$ = 3.3$^{E-9}$ $p$ < 0.05, modified Bonferroni test, $n$ are listed below the x-axis). Data underlying this figure can be found in S6 Data. FRET, Förster Resonance Energy Transfer; l-LNv, large ventral lateral neuron; PDF, pigment dispersing factor; PDFR, pigment dispersing factor receptor.

the respecification of the PDFR in the l-LNvs during sleep restriction. Indeed, the sensitivity of the l-LNvs to PDF was restored in well-rested mature adults by overexpressing *UAS-nejire$^{WT}$*. Conversely, the respecification of the PDFR to the l-LNvs during sleep restriction was blocked by *UAS-nejire$^{RNAi}$* (Fig 6D). Together, these data reveal that conditional PDFR expression in l-LNvs shares common mechanisms in both young flies and mature adults.

Finally, we asked whether similar mechanisms were used by young and mature adults for the activation of *nejire*. To identify cell surface receptors that might interact with *nejire*, we once again consulted a database of genes known to be enriched in the LNvs [50]. We then conducted a targeted RNAi screen to evaluate PDF sensitivity in young flies and mature adults following sleep restriction (Figs 6E and 6F and S6). Given that we only evaluated 1 RNAi line to evaluate each receptor, these results should be viewed cautiously.

Nonetheless, these data suggest the possibility that *DopEcR*, *Dop1R2*, and *InR* may play a role in the respecification of the PDFR in both young flies and mature adults during sleep restriction.

## Discussion

In *Drosophila*, PDF differentially coordinates the activity of diverse neuronal groups to optimize behavioral output with the prevailing environmental conditions [16,51]. One of the most successful topics in circadian neurobiology today has been the use of circuit mapping to decipher the logic used by the clock to regulate such rhythmic behavior [10,11,13,52,53]. Indeed, a number of studies highlight the role of PDF in coordinating the activity of downstream circuits [16,41,54]. In this manuscript, we replicate previous finding that the Pdfr is not normally expressed in the wake-promoting l-LNvs of healthy adults [13,29,30]. However, we now show that high sleep pressure quickly reprograms the l-LNvs to express the Pdfr. The addition of a new signaling input into the circuit, through expression of the Pdfr in the l-LNvs, is associated with increased waking and early mating success. Importantly, these data demonstrate that the constellation of neurons that express Pdfr is not constant and points to a novel type of plasticity that can be used by the clock to coordinate behavioral output.

A growing number of studies indicate that sleep regulatory mechanisms are plastic and can be harnessed to match an individual's sleep need with environmental demands [21,46,55]. Although Hebbian and synaptic plasticity modulate circuit function in a variety of contexts, these forms of plasticity may not be well suited to sculpt the balance of sleep and wake-promoting circuits at specific times of day [56]. In contrast, receptor respecification is a form of plasticity that may allow an individual to engage in adaptive waking behaviors at optimal circadian times while still allowing the animals to obtain needed sleep at other times [19]. Indeed, our data indicate that the PDFR is transiently expressed in wake-promoting clock neurons during the first approximately 48 h of adult life when sleep drive is high. The associated increase in waking is confined to a small portion of the circadian day and supports mating success and mating competition. In contrast, the response properties of the l-LNvs to the global wake-promoting transmitters Oa and DA remains unchanged [57]. Interestingly, when sleep is disrupted in 5-day-old adults, the PDFR is once again expressed in the l-LNvs. Thus, targeted receptor respecification may be an effective strategy that can be used to support important, species-specific behaviors during conditions of high sleep drive without substantially disrupting the ability of the animal to obtain needed sleep.

Our data indicate that there is a strong relationship between sleep drive and the respecification of the PDFR in a subset of clock neurons. That is, while the l-LNvs are unresponsive to PDF in mature adults [30], the l-LNvs display robust responses to PDF following sleep deprivation, sleep restriction, and prolonged starvation. Importantly, no changes in the response properties of the l-LNvs were observed when the animals were exposed to the mechanical stimulus in the absence of sleep restriction. Interestingly, the response properties of the l-LNvs was not visible until the second day of sleep restriction indicating that low amounts of sleep drive are not sufficient to respecify the PDFR. Consistent with this hypothesis, short durations of starvation induce episodes of waking that are not compensated by a sleep rebound [58] and do

not change the response properties of the l-LNvs to PDF. In contrast, after approximately 20 h of starvation, a time when flies begin to display a sleep rebound, the l-LNvs begin to respond to PDF. These data suggest that the PDFR may be respecified in the l-LNvs to, for example, help sleepy animals stay awake long enough to support a basal level of foraging. Indeed, knocking down the *Pdfr* in clock neurons results in a larger sleep rebound following sleep restriction. Increased sleep in many circumstances may be maladaptive since it would likely limit the opportunity to forage or mate [59]. Together, these data support the hypothesis that the PDFR is expressed to assist waking behaviors during conditions of high sleep drive.

Given that high sleep drive can negatively impact male sexual behavior (Chen and colleagues [55]), it is curious that the PDFR is not typically expressed in the l-LNvs of healthy adults. However, previous studies have shown that genes that confer resilience to specific environmental challenges can be deleterious in other circumstances (Donlea and colleagues [45]). Indeed, the exogenous expression of PDFR in the l-LNvs during adulthood reduced survival during prolonged starvation. These data suggest that the normal down-regulation of PDFR expression in l-LNvs of healthy adults may be advantageous in that it removes potentially excessive behavioral drives that could deplete valuable resources. Indeed, genetically preventing PDFR expression in l-LNvs during starvation extended survival.

Although sleep drive does not change the response properties of the l-LNvs to DA, our data suggest that changes in dopaminergic tone may play a role in the respecification of the PDFR in the l-LNvs. Specifically, knocking down specific DA receptors in the l-LNvs prevents the respecification of the PDFR in both young flies and sleep-restricted 5-day-old adults. Although the precise dopaminergic neurons have not yet been identified, the PPL2 dopaminergic neurons project to the l-LNvs to promote wakefulness [40,41] and may play a role in the expression of the PDFR in l-LNvs. In addition to DA receptors, our data identify a role of the transcription factor *nejire* (cAMP response element-binding protein) in promoting the expression of the PDFR during conditions of high sleep drive. Interestingly, *nejire* plays a role in circadian function where it has been suggested to allow cross-talk between circadian transcription and the transcriptional regulation of other important processes such as sleep, metabolism, and memory formation [60,61].

Previous studies have shown that activity-dependent respecification of receptors in mammals can occur in adult neurons in response to >1 week of sustained increases in sensory activity [18,19]. The most common forms of respecification alter the polarity of the synapse to alter the function of the circuit [19]. Our data suggest an additional type of respecification in which an input pathway into a circuit can be turned on and off, without changing the sign of the synapse (excitatory/inhibitory). Presumably, turning on an input pathway may be a rapid first step to balance the impact of sustained activity in opposing circuits (e.g., sleep versus wake). However, enhancing the activity of a circuit may create a positive feedback loop, which can destabilize the system and lead to adverse consequences. Indeed, while the respecification of the PDFR in the l-LNvs can improve mating success during high sleep drive, it also results in early lethality during starvation. Understanding how sleep drive modulates respecification plasticity in other sleep regulatory circuits may provide critical insight into the role that sleep plays in maintaining adaptive behavior in an ever changing environment.

## Materials and methods

### Flies

Flies were cultured at 25˚C with 50% to 60% relative humidity and kept on a diet of yeast, dark corn syrup, and agar. Newly eclosed males were collected and entrained 4 to 7 days in a 12-h:12-h light:dark (LD) cycle, unless otherwise specified. RNAi stocks were obtained from

VDRC and TRiP stock centers. $DopEcR^{RNAiJF03415}$, $Dop1R1^{RNAiHM04077}$, $Dop1R2^{RNAiHMC06293}$, $D2R^{RNAiHMC02988}$, $InR^{RNAiHMS03166}$, $NPFR^{RNAiJF01959}$, $Oamb^{RNAiJF01673}$, $TkR86C^{RNAiJF02160}$, $Oa2^{RNAiHMJ22156}$, $Cry^{RNAiJF01880}$, $Mael^{RNAiHMS00102}$, $dimm^{RNAiHMS01742}$, $mamo^{RNAiHMC03325}$, $cac^{RNAiHM04020}$, $achintya^{RNAiHMS01127}$, $SuUR^{RNAiGL01080}$, $nejire^{RNAihp12}$, $nejire^{RNAihp12.3}$. Other stocks used were $c929(dimm)$-GAL4; PDF-GAL4; $lLNv^{GRSS000645}$-GAL4 (G. Rubin, H. Dione, A. Nern); $UAS$-$nejire^{wt-V5}$, $pdf^{01}$, $w^{1118}$ [37]. All other UAS lines and GAL4 lines have been described previously: $Pdfr$-null mutant ($Pdfr^{5304}$); $UAS$-$Pdfr^{wt}$; w; $UAS$-$Epac1camps50A$ [30], w, Pdf-GAL4(M) and UAS-$Pdfr^{RNAi-42508}$ [62]. c929-GAL4; cry-GAL80/UAS-GFP flies and P [acman] pdfr-myc70 flies [29] were used for immunolabeling.

## Sleep

Sleep was measured as described previously [26]. In short, individual flies were placed into approximately 65 mm tubes, which were then placed into Trikinetics Drosophila Activity Monitoring System (www.Trikinetics.com, Waltham, Massachusetts). Locomotor activity was monitored using an infrared beam and was assessed using 1-min time bins. Sleep has been defined as periods of quiescence lasting 5 min or longer [26].

## Mating success

Mating success assay consisted of putting 1 virgin female in a vial with a single male of the genotype to be evaluated. Each pair of flies was placed into a vial at ZT4 on the day of eclosion, and then the male was removed at ZT24. Mating success was determined days later through visual inspection of viable offspring (pupae, larvae, etc.). Females from vials that produced no offspring were subsequently provided several males to test for her reproduction viability.

## Mating competition

A mating competition assay was also carried out using 2 males who compete for 1 female. In each tube, 1 white-eyed male and 1 red-eyed male of varying PDFR levels competed to mate with a white-eyed female. The 2 competing males were added to a vial simultaneously with a mature virgin female, just prior to the WMZ (ZT10) and left in the vial until the end of WMZ (ZT12). Successful mating of the red-eyed male was determined by female progeny with eye color. Twenty competitions were set up for each genotype and repeated 3 times. Only competitions resulting in progeny were used for analysis.

## Sleep restriction

Disruption of sleep was performed similarly as previously described [38,63]. Flies were placed into individual 65 mm tubes and a sleep-nullifying apparatus (SNAP), which mechanically disrupted sleep for 1 min every 15 min for 24 to 48 hours, which both reduced and fragmented sleep. For sleep deprivation, the SNAP was activated once every 8 s for the duration of the experiment. Sleep homeostasis was calculated for each individual as a ratio of the minutes of sleep gained above baseline during the 48 h of recovery divided by the total minutes of sleep lost during 12 h of sleep deprivation.

## Starvation

For starvation experiments, adult flies loaded into Trikinetics tubes containing 1% agar, which provides water but not nutrients. Flies' behavior was monitored until being evaluated for imaging or for survival experiments. The duration of starvation is noted in the text.

## Physiology

Methods generally followed those of Klose and colleagues [31]. Flies were removed from DAM monitors, and glass tubes were placed on ice for approximately 5 min. Three to 4 flies were pinned onto a sylgaard dissection dish and were dissected in cold calcium-free HL3 (Stewart and colleagues [64]). Dissected brains were transferred onto a polylysine treated dish (35 3 10 mm Falcon polystyrene) containing 3 ml of 1.5 mM calcium HL3. Two to 4 brains were assayed concurrently, typically a mutant line and its genetic controls. Image capture and x,y,z stage movements were controlled using SLIDEBOOK 5.0 (Intelligent Imaging Innovations, Denver, CO, USA), which controlled a Prior H105Plan Power Stage through a Prior ProScanII. Multiple YFP/CFP ratio measurements were recorded in sequence from region of interest (ROI) in each hemi-segment of each brain in the dish. Each ROI comprised 2 to 4 l-LNvs. Following baseline measurements, 1 ml of saline containing various concentrations of either PDF, DA, or OA (Sigma-Aldrich, St. Louis, MO, USA) was added to the bath (dilution factor of 1/4). We tested normality in the data using the Shapiro–Wilk test in SigmaPlot (Systat Software, San Jose, CA, USA); maximum amplitude values were used to perform ANOVA analyses followed by post hoc Tukey tests.

## Immunocytochemistry

Whole flies were fixed in 4% PFA for several hours, and brains were then dissected in ice-cold PBS and incubated overnight with the following primary antibodies: mouse anti-PDF, (5F10, 1,10 dilution, Hybridoma Bank, University of Iowa), chicken anti-myc (GFP-1020; 1:1,000), and anti-GFP. Secondary antibodies were Alexa 488 and 633 conjugated at a dilution 1:200. Brains were mounted on polylysine-treated slides in Vectashield H-1000 mounting medium. Confocal stacks were acquired with a 0.5-μm slice thickness using an Olympus FV1200 laser scanning confocal microscope and processed using ImageJ.

## Statistics

All comparisons were done using a Student $t$ test or, if appropriate, ANOVA and subsequent planned comparisons using modified Bonferroni test unless otherwise stated. All statistically different groups are defined as $^*p < 0.05$.

## Supporting information

**S1 Fig.** (A, B) Response of l-LNvs to DA and Oa in *Pdf-GAL4>UAS-Epac1* flies from day 0 to day 5+ ($n$ = 4–15 hemi-segments per age, ANOVA $F_{[5,36]}$ = 6.08, $p$ = 0.96 and ANOVA $F_{[5,54]}$ = 8.93, $p$ = 0.90, respectively). (C) Normalized FRET ratio in s-LNvs before and during PDF exposure on day 0 ($n$ = 6) and day 5 ($n$ = 7). (D) PDF response amplitude in s-LNvs on day 0 to day 5+ (ANOVA $F_{[5,76]}$ = 13.31, $p$ = 3.75$^{E-9}$ $n$ = 8–20 hemi-segments per age). $^*p < 0.05$, modified Bonferroni test. Data underlying this figure can be found in S7 Data. DA, dopamine; FRET, Förster Resonance Energy Transfer; l-LNv, large ventral lateral neuron; Oa, octopamine; PDF, pigment dispersing factor; s-LNv, small ventral lateral neuron.
(TIF)

**S2 Fig.** (A) Immunohistochemistry for PDF and GFP reveals the expression of GFP in the l-LNvs of *c929-GAL4/UAS-gfp* flies but not in the brains of *c929-GAL4/UAS-gfp; Cry-Gal80* flies. (B) *Pdfr$^{5504}$; PDF>/UAS-Pdfr$^{WT}$* flies exhibit more waking during the WMZ than *Pdfr$^{5304}$; Pdf-GAL4/+* and *Pdfr$^{5304}$;UAS-Pdfr$^{WT}$/+* parental controls (ANOVA $F_{[2,91]}$ = 4.63, $p$ = 0.01 $n$ = 41–64 flies/genotype) flies. Data underlying this figure can be found in S8 Data. GFP, green fluorescent protein; l-LNv, large ventral lateral neuron; PDF, pigment dispersing

factor; WMZ, wake maintenance zone.
(TIF)

**S3 Fig. CD8 GFP expression using l-LNv–specific split Gal4 driver GRSS000645.** (A) CNS with overlay traced reveals cell bodies and optic lobe projections of l-LNvs in the left hemi-segment of a brain. (B) Four cell bodies and projections of l-LNvs of right hemi-segment. Z-stack projections with 2 μm steps. Scale bar: 15 μm. GFP, green fluorescent protein; l-LNv, large ventral lateral neuron.
(TIF)

**S4 Fig.** (A) Sleep was reduced in both $pdf^{01}$ mutants and $w^{1118}$ genetic controls during the first 18 h of starvation (data presented as change from baseline; ($n = 20$–22 flies/genotype, $p > 0.05$). (B) During the first 18 h of starvation, waking activity was significantly lower in $pdf^{01}$ mutants compared to $w^{1118}$ controls ($p < 0.05$). (C) Kaplan–Meier analysis reveals % survival during starvation in $pdf01$ ($n = 25$) flies and w1118 ($n = 24$) controls ($\chi^2 = 6.20$, df = 1, $p = 0.01$). Data underlying this figure can be found in S9 Data. n.s., not significant.
(TIF)

**S5 Fig.** (A) Sleep (minutes) during 48 h of starvation in *Dcr2; c929*-GAL4/UAS-*pdfr*$^{RNAi}$ flies ($n = 24$), *Dcr2; c929-GAL4/+* ($n = 18$), and +/UAS-*Pdfr*$^{RNAi}$ ($n = 32$) control flies. ZT, Zeitgeber time.
(TIF)

**S6 Fig.** (A) The amplitude of l-LNv responses to DA on day 0 in *Pdf-GAL4>UAS-Epac1* flies coexpressing RNAi lines for the depicted transcription factors (ANOVA $F_{[8,92]} = 1.04$, $p = 0.42$; $n$ is as indicated beneath each bin). (B) The amplitude of s-LNvs responses in *Pdf-GAL4>UAS-Epac1* flies coexpressing RNAi lines for the depicted transcription factors neurons on day 0 (ANOVA $F_{[8,112]} = 9.36$, $p = 1.19^{E-9}$). (C) The amplitude of s-LNvs responses to DA on day 0 in *Pdf-GAL4>UAS-Epac1* flies coexpressing RNAi lines for the depicted cell surface receptors (ANOVA $F_{[10,108]} = 0.79$, $p = 0.63$; $n$ is as indicated beneath each bin). (D) The amplitude of l-LNvs responses to DA following sleep restriction in 5-day-old *Pdf-GAL4>UAS-Epac1* flies coexpressing RNAi lines for the depicted cell surface receptors (ANOVA $F_{[10,159]} = 0.42$, $p = 0.94$; $n$ is as indicated beneath each bin). Data underlying this figure can be found in S9 Data. DA, dopamine; l-LNv, large ventral lateral neuron; PDF, pigment dispersing factor; RNAi, RNA interference; s-LNv, small ventral lateral neuron.
(TIF)

**S1 Data. (A, B)** Sleep in minutes/hours for 0-, 1-, and 3-day-old flies maintained on a 12:12 LD schedule. **(C–E)** FRET ratio measurements in *Pdf-GAL4>*UAS-*Epac1* flies in response to DA, Oa, and PDF. **(F)** The amplitude of l-LNv responses to PDF. DA, dopamine; FRET, Förster Resonance Energy Transfer; LD, light:dark; l-LNv, large ventral lateral neuron; Oa, octopamine; PDF, pigment dispersing factor.
(XLSX)

**S2 Data. (A)** Sleep in minutes/hours in Pdfr$^{5504}$, Cs, Pdfr$^{5504}$; c929-GAL4; UAS-Pdfr (rescue, green), Pdfr$^{5504}$; UAS-Pdfr/+, Pdfr$^{5504}$; c929-GAL4/+; Cry-Gal80/+, and Pdfr$^{5504}$; c929-GAL4/UAS-Pdfr; Cry-Gal80.
(XLSX)

**S3 Data. Role for PDFR in l-LNvs in mating success. (A–C)** % of vials with offspring for *Cs*, *Dcr2; c929*-GAL4/UAS-*Pdfr*$^{RNAi}$, *Dcr2; c929-GAL4/+*, +/UAS-*Pdfr*$^{RNAi}$, Pdfr$^{5504}$; c92-GAL4; UAS-*Pdfr*$^{WT}$, Pdfr$^{5504}$; UAS-*Pdfr*$^{WT}$/+, *Pdfr*$^{5504}$; c929-GAL4/+; CryGal80/+, and *Pdfr*$^{5504}$; c929-

GAL4/UAS-*Pdfr*; *Cry*-Gal80. **(D)** Mating competition assay scheme on day 1. **(E–G)** % of vials with red eyes.
(XLSX)

**S4 Data. (A)** Daytime sleep in *l-LNv-GAL4>; UAS-Pdfr^{WT}/+, l-LNv-GAL4/+, UAS-Pdfr^{WT}/+ Dcr2; c929*-GAL4/UAS-*Pdfr, c929*-GAL4/+, and UAS-*Pdfr*/+ flies **(C)** % of surviving flies each hour during starvation in *l-LNv-GAL4>; UAS-Pdfr^{WT}/+, l-LNv-GAL4/+, UAS-Pdfr^{WT}/+.
(XLSX)

**S5 Data. (A)** Traces of normalized FRET ratio during PDF application in l-LNvs from starved and fed *Pdf-GAL4>UAS-Epac1* flies. **(B)** The amplitude of l-LNv responses to PDF in 5-day-old *Pdf-GAL4>UAS-Epac1* flies following 21–24 h of starvation. Data are shown for 8 h bins. **(C)** Sleep in minute/hour in *Cs* flies maintained on a 12:12 LD schedule during sleep restriction day 2 and during recovery. **(D)** Traces of normalized FRET ratio during PDF application in l-LNvs recorded from *Pdf-GAL4>UAS-Epac1* flies during sleep restriction sleep deprivation and bang. **(F)** The amplitude of l-LNv responses to PDF in l-LNvs during baseline, sleep restriction, and recovery. **(G)** Sleep rebound expressed as a difference with baseline sleep in *Dcr2; c929-GAL4/ UAS-Pdfr^{RNAi}, Dcr2; c929-GAL4/+,* and *UAS-Pdfr^{RNAi}/+* flies. FRET, Förster Resonance Energy Transfer; LD, light:dark; l-LNv, large ventral lateral neuron; PDF, pigment dispersing factor.
(XLSX)

**S6 Data. (A)** The amplitude of l-LNv responses to PDF on day 0 in *Pdf-GAL4>*UAS-*Epac1* flies crossed to *UAS-RNAi* lines of the labeled transcription factors. **(B)** Trace of PDF responses in *Pdf-GAL4>UAS-Epac1* during baseline, in sleep-restricted flies, and *Pdf-GAL4>UAS-Epac1* flies expressing *UAS-nej^{RNAi}*. **(C, D)** The amplitude of l-LNv responses in *Pdf-GAL4>UAS-Epac1/UAS-nej^{WT}* and age-matched *Pdf-GAL4>UAS-Epac1* flies. **(E)** The amplitude of l-LNv responses to PDF *Pdf-GAL4>UAS-Epac1* flies crossed to *UAS-RNAi* lines of the listed cell surface receptors. **(F)** The amplitude of l-LNv responses to PDF in sleep restricted. l-LNv, large ventral lateral neuron; PDF, pigment dispersing factor.
(XLSX)

**S7 Data. (A, B)** Response of l-LNvs to DA and Oa in *Pdf-GAL4>UAS-Epac1* flies from day 0 to day 5+. **(C)** Normalized FRET ratio in s-LNvs before and during PDF exposure on day 0 and day 5. **(D)** PDF response amplitude in s-LNvs on day 0 to day 5+. DA, dopamine; FRET, Förster Resonance Energy Transfer; l-LNv, large ventral lateral neuron; Oa, octopamine; PDF, pigment dispersing factor; s-LNv, small ventral lateral neuron.
(XLSX)

**S8 Data. (B)** Sleep during the waking during the WMZ in *Pdfr^{5304}; Pdf-GAL4/+ Pdfr^{5304}; UAS-Pdfr^{WT}/+* and *Pdfr^{5504}; PDF>/UAS-Pdfr^{WT}* flies. WMZ, wake maintenance zone.
(XLSX)

**S9 Data. (A)** Sleep in *pdf^{01}* mutants and *w^{1118}* flies during the first 18 h of starvation (data presented as change from baseline. **(B)** Waking activity during the first 18 h of starvation. **(C)** Surviving flies each hour during starvation.
(XLSX)

**S10 Data. (A)** The amplitude of l-LNv responses to DA on day 0 in *Pdf-GAL4>UAS-Epac1* flies coexpressing RNAi lines for the listed transcription factors. **(B)** The amplitude of s-LNvs responses in *Pdf-GAL4>UAS-Epac1* flies coexpressing RNAi lines for the listed transcription factors neurons on day 0. **(C)** The amplitude of s-LNvs responses to DA on day 0 in in *Pdf-GAL4>UAS-Epac1* flies coexpressing RNAi lines for the depicted cell surface receptors. **(D)**

The amplitude of l-LNvs responses to DA following sleep restriction in 5-day-old *Pdf-GAL4>UAS-Epac1* flies coexpressing RNAi lines for the listed cell surface receptors. DA, dopamine; l-LNv, large ventral lateral neuron; RNAi, RNA interference; s-LNv, small ventral lateral neuron.
(XLSX)

## Acknowledgments

We thank Gerald Rubin and Paul Taghert for sharing reagents and flies. We also thank Stefan Dissel for help throughout this project, Lijuan Cao for help with immunohistochemistry, and Krishna Melnattur for comments.

## Author Contributions

**Conceptualization:** Markus K. Klose, Paul J. Shaw.

**Data curation:** Markus K. Klose, Paul J. Shaw.

**Formal analysis:** Markus K. Klose, Paul J. Shaw.

**Funding acquisition:** Paul J. Shaw.

**Investigation:** Markus K. Klose, Paul J. Shaw.

**Methodology:** Markus K. Klose, Paul J. Shaw.

**Project administration:** Paul J. Shaw.

**Resources:** Paul J. Shaw.

**Supervision:** Paul J. Shaw.

**Validation:** Markus K. Klose, Paul J. Shaw.

**Writing – original draft:** Markus K. Klose, Paul J. Shaw.

**Writing – review & editing:** Markus K. Klose, Paul J. Shaw.

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
