## [Editor Report · Decision Letter 0]

3 Sep 2020

Dear Dr Shaw, 

Thank you for submitting your manuscript entitled "Sleep-drive reconfigures clock circuitry to regulate adaptive behavior" for consideration as a Research Article by PLOS Biology.

Your manuscript has now been evaluated by the PLOS Biology editorial staff as well as by an academic editor with relevant expertise and I am writing to let you know that we would like to send your submission out for external peer review.

Please re-submit your manuscript within two working days, i.e. by Sep 07 2020 11:59PM.

Kind regards,

Lucas Smith, Ph.D.,

Associate Editor

PLOS Biology

---

## [Decision Letter · Decision Letter 1]

26 Oct 2020

Dear Dr Shaw,

Thank you very much for submitting your manuscript "Sleep-drive reconfigures clock circuitry to regulate adaptive behavior" for consideration as a Research Article at PLOS Biology. Your manuscript has been evaluated by the PLOS Biology editors, an Academic Editor with relevant expertise, and by several independent reviewers.

As you will see from their detailed responses, the reviewers are enthusiastic about the study in principle, but raise a number of concerns that would need to be addressed to bolster the conclusions drawn in this study. In particular, a concern was raised with the fact that the potential promiscuity of PDFR makes it hard to know whether the effects are acting via PDF or other signals. The reviewers also questioned the reliance on the C929 driver, which is expressed in regions outside the lLNv, thereby making it difficult to say whether the effects are lLNv-specific, and would like additional mechanistic insights.

In light of the reviews (below), we will not be able to accept the current version of the manuscript, but we would welcome re-submission of a much-revised version that takes into account the reviewers' comments. We cannot make any decision about publication until we have seen the revised manuscript and your response to the reviewers' comments. Your revised manuscript is also likely to be sent for further evaluation by the reviewers.

We expect to receive your revised manuscript within 3 months, however please let us know if you need more time to complete the revision, as we would be happy to grant an extension. 

**IMPORTANT - SUBMITTING YOUR REVISION**

*Re-submission Checklist*

*Published Peer Review*

*PLOS Data Policy*

*Blot and Gel Data Policy*

Sincerely,

Lucas Smith, Ph.D.,

Associate Editor,

lsmith@plos.org,

PLOS Biology

REVIEWS:

Reviewer's Responses to Questions

PLOS authors have the option to publish the peer review history of their article (what does this mean?). If published, this will include your full peer review and any attached files.

Reviewer #1: No

Reviewer #2: No

Reviewer #3: No

Reviewer #4: No

Reviewer #1: In this study, Klose and Shaw describe a striking phenomenon in which an important GPCR for circadian timekeeping (PdfR) is expressed transiently within wake promoting clock neurons (the l-LNvs) during early adulthood to promote late day wakefulness during a developmental window characterized by high levels of sleep. Remarkably, they show that the expression of PdfR is reinstated during times of extended sleep disruption. The authors also examine the fitness implications for this phenomenon and address the relationships between sleep disruption and l-LNv PdfR expression in detail. They also go on to implicate genes mediating the context dependency of PdfR expression in the l-LNvs. These results are highly significant and of great interest. The following points, all modest or minor, might be considered by the authors to improve this very interesting contribution to the field:

1.) What about the PDF neuropeptide? PdfR is a type II GPCR, a class of receptors known for their promiscuity. Are the effects of PdfR signaling in the l-LNvs during times of high juvenile sleep drive and during prolonged sleep disruption mediating PDF signals or signals from some other peptide? It's surprising not to see the Han5304 experiments repeated for Pdf01 mutants. The results of this analysis would be interesting either way and the central conclusion would not depend on the outcome, but the absence of Pdf01 data here produces a bit of an itch that could be easily scratched.

2.) The authors should provide a more detailed rationale for focusing on the 2-h time-window just before dusk as a time corresponding "high sleep-drive." This might confuse some readers, as this is the window in which flies show anticipatory evening activity. Readers might therefore be predisposed to expect low sleep drive here. The idea that this is specific to sleepy young flies could be clarified.

3.) The description of the methods for gauging mating success should be more detailed. Without more detail is it difficult to gauge the quality of the experimental design and the appropriateness of the statistical analysis. 

4.) There are no error bars shown on figure depicting mating success data (3A-C), where these experiments replicated?

5.) Although the survival results depicted in Figs. 4C and D are quite striking and clear, a Kaplan-Meier survival analysis should be done for the statistical analysis on these data.

6.) In Fig. 5D, the baseline sleep has clear MYC signal, which is not is what is reported in the results. Is this a labeling error (i.e., are PDF and and MYC labels switched)?

7.) The data shown in supplemental figure S3 seems critical and should be placed in the main figures (Fig. 5).

8.) On Page 2, the four citations (10-13) provided for the notion that circadian timekeeping is a distributed network property neglect two important studies at the expense of a review and a study of questionable relevance (Yao et al. 2014 Science Vol. 343, p. 1516 and Yao et al. 2016 Cell Reports Vol. 13, p.2873).

9.) The use of the split GAL4 specific for the l-LNvs is fantastic, but it's used only in a subset of the experiments. The authors should consider using it alongside all c929-GAL4 experiments, as c929 is widely expressed throughout the brain.

10.) Page 8, line 11, "factor" should probably be "factors."

11.) Page 9, line 1: I don't think it's accurate to call PDF "the main output of the clock" at this point (it also brings the itch of the absence of PDF peptide data back).

12.) Page 19, line 2: "~20" should be "~20h."

13.) I'm not entirely sure about the term "respecification" here. Might there be a better term or phrase?

14.) "Data Table S1 does not match the figures presented in the manuscript.

15.) The text on the histogram in Fig. 2D is difficult to read.

Reviewer #2: This is an interesting manuscript that presents evidence for a dynamic expression of Drosophila PDFR in a specific group of circadian neurons - the lLNvs - known to regulate arousal. PDFR expression in adults is limited to the first couple of days following eclosion, when sleep is high. Also, PDFR expression can be reactivated in case of sleep restriction. PDFR appears to ensure wakefulness in the evening, and thus improves mating in young adult and survival from starvation. The last figure of the paper presents a mini-screen for genes that might contribute to PDFR signaling. This part of the manuscript is quite preliminary, but otherwise, overall, this appears to be an important and for the most part solid study. 

My main suggestion to improve this manuscript would be to more clearly establish that the genes identified in figure 6 participate in PDFR signaling and sleep regulation. Do these genes indeed affect sleep similarly to PDFR, do they genetically interact with PDFR?. Also, I do not understand why downregulation of dopamine receptors decrease PDFR responses, but oddly do not reduce sensitivity of the lLNvs to dopamine. This needs to be explained. 

I also have a couple of concerns about specific experiments:

1) Is rescue of fig 2b statistically significant? Why is it much weaker than with c929-GAL4? PDF-GAL4 is a strong driver

2) I would suggest that the authors overexpress PDFR in lLNvs of wild-type flies with c929-GAL4 and PDF-GAL4 to ensure that this does not result in gain of functions that could confound the rescue experiments. 

3) Why do control flies sleep so little in the RNAi experiments on fig. 2E? Are these flies defective for juvenile sleep regulation, or is there considerable variations between genetic backgrounds? This brings me to wonder how carefully genetic backgrounds are controlled. Are all flies backcrossed to CS? This was not explained in the material and methods or in the main text. 

Minor comments:

1) Is "receptor respecification" a commonly used term? It would be helpful to define this as it is used throughout the manuscript

2) The first sentence of the last paragraph of p.2 should be rephrased, as it sounds as if sleep is an environmental cues. It is also not clear to me how sleep synchronizes central and peripheral clocks

4) The first sentence of p.3 needs references

5) My understanding is that only about half of circadian neurons express PDFR, based on work by the Shafer and Taghert labs (Im et al., 2010, 2011 and Yao and Shafer 2014). The relevant statement at the end of the introduction should be corrected. 

6) The H on figure 1 is missing.

7) I would indicate on page 4 that there is a noticeable reduction in PDFR sensitivity in sLNv

8) Cry-gal80 is only expressed in CRY+ neurons. The relevant statement at the bottom of p.4 needs to be corrected. 

9) On p.4, the authors say that they hypothesized PDFR would be there to promote wakefulness in the evening. While indeed this was the case, the authors could explain better why they thought this would be the case. It seems to me PDFR could have been there to promote sleep. 

10) Panels shown on figure 3 are also present on figure S2 

11) Second paragraph, p.5, correct "PDF" to "PDFR"

12) Figure 5E. PDf amplitude is not a clear label for the y-axis. Also, it would be clearer to spell out day 1 and day 2 rather then just having 1 or 2 on the x-axis

Reviewer #3: The authors present convincing data that Pdfr expression and receptivity to PDF changes with age in the large LNv, and that perturbations in adult sleep or feeding can induce Pdfr expression in the lLNv. The authors also find that the behavioral effects of loss of Pdfr appear different in young flies than phenotypes previously reported in older adults.

Major Comments:

1. It's not clear whether the physiologically relevant changes in PDFR expression/ PDFR signaling are mainly limited to the large LNv or if other cell groups may contribute. The authors rely primarily on C929-GAL4 to argue for function in the lLNv group, but C929 is expressed outside of the lLNv. The use of cryGAL80 with C929 is helpful but may not limit function to the lLNv given that it blocks expression in other clock neuron groups. Instead, pdfGAL80 would be a much cleaner reagent. 

2. In addition, the authors describe a split driver that is specific to the lLNv—the expression pattern and specificity of this line should be documented and functional analysis performed to confirm pdfr function in the lLNv.

3. Young pdf5304 flies sleep more and have less success at mating. If sleep and mating success are related, would other types of wake or sleep promotion (pharmacological, thermogenetic) in young wild type flies also affect mating chances? Also, are pdf5304flies healthy, or can their reduced success at mating/increased sleep be explained as a sickness phenotype? 

Other specific comments:

1. Experimental details for assaying Day 0 behavior were included for Figure 2 but not Figure 1A-B. Please clarify method. 

2. For Figure 1G, it would be helpful to compare representative images between Day 0 and Day 5. Are there any other differences observed in PDFR expression between these ages? The findings in S1c-d suggest PDFR expression in the sLNv might decrease as well. If there are notable increases/decreases in PDFR levels in other groups, this should be quantified.

3. Statistical significance indicated in some graphs/panels but not others (eg. Figure 3A-C). Please also clarify statistical method used for Figure 3A-C. 

4. Previous reports have demonstrated a role for PDF/PDFR in adult sleep regulation (Parisky 2008, Chung 2009), including increased sleep in light phase in PDF/PDFR mutants. The authors should relate their current findings to these published data. A comparison in phenotypes between Day 0 and Day 5 for your backcrossed strains would be informative. 

5. The methods relating to starvation related assays (Fig 4C,D; Fig 5A; Fig S3A) were not clearly described.

6. The format of Figure S3a is confusing given that some of the timepoints have an n of 0, thus PDF amplitude was not determined. Additional timepoints should be added and/or data should be binned differently.

7. The methods should more clearly describe the differences between the 'sleep deprivation' and 'sleep restriction' protocols (Fig 5C).

8. Figure 5D, it is difficult to identify PDF+ MYC- large LNv in the baseline images shown. Higher resolution images should be provided with labels to indicate large LNv, ideally comparing a full cluster of large LNv for both baseline and sleep-restricted conditions.

9. neijere has previously been implicated in regulating CLK-CYC transcription activity- does nej RNAi affect pdfGAL4/ epac expression? 

10. It is not clear which UAS-pdfr RNAi strain(s) were used.

11. "Receptor respecification is a form of plasticity that, like Hebbian and homeostatic plasticity, may be employed to alter circuit function in response to changing environmental demands [16]" 

Can the authors elaborate more on receptor respecification? I find the term a bit confusing - it suggests that one receptor is changed, respecified, to another receptor. But here it seems that a receptor appears in a circuit where it hasn't been previously observed. Also, the appearance of a new receptor should coincide with a change in presynaptic neurotransmitters. Can the authors speculate on which presynaptic inputs have changed? 

12. After the sleep restriction protocol that restricts fly sleep to bouts of 15 minutes, flies do not show increased rebound sleep. Is their sleep architecture (bout lengths, bout numbers) also unchanged? 

Also, during the one minute sleep deprivation, how many times did the SNAP device move? 

13. Fig 4C,D - survival curves are usually represented as Kaplan Meier plots and tested for significance using a logrank test. Can the authors reanalyze their data this way? 

Reviewer #4: Comments on "Sleep drive reconfigures clock circuitry…"

This is a promising paper with a set of interesting ideas woven together. It piggy backs onto the outstanding work of Kayser et al. (Sehgal lag) and more recently Matt Kayser from his own lab. They showed that very young flies, days 1-3 approximately after eclosion, sleep considerably more than more mature adults - like human babies. And the more recent work explores the molecular underpinnings of the initial observations. This paper takes off from these observations and finds that the expression of the famous PDF receptor (PDFR) in a well-studied group of circadian neurons is regulated in a similar fashion, namely, PDFR is expressed in the large ventral lateral neurons (l-LNvs) in very young adult flies but then disappears from these cells by day 5. This receptor is known to be expressed in many places in the fly brain including in the l-LNv's neighboring cells, the small ventral lateral neurons (s-LNvs), where it is present all the time in adult flies. The paper then goes on the explore the role of this receptor and how its expression is regulated. They argue that it is expressed in response to a number of stresses including (a particular version of) sleep deprivation, and they identify a transcription factor likely responsible for its synthesis as well as some receptors expressed in the l-LNvs that may be upstream of transcription factor activation.

Comments:

1) From the outset the paper is difficult to follow. The framing and beginning of the paper leads the reader to believe that PDFR expression in l-LNvs must contribute to sleep (since these very young flies sleep more), and yet it does the opposite. Indeed, these neurons have been shown to contribute to daytime activity and therefore would be expected to inhibit sleep. The presence of the receptor causes them to do this only more so it seems to me. A clear definitive sentence at the end of the first paragraph of the Results saying something like this and leading into paragraph 2 would be helpful.

2) The paper relies heavily on the han (PDFR) mutant strain and I think a single RNAi. Also, the key figure 2A should be done in parallel with the pdf01 mutant strain. The results should be very similar, no? And Figure 2A is missing the data from ZT0-ZT4. Nonetheless, it looks like the WT has morning anticipation here but not in Fig. 1. How come? And I'd like to see the % sleep change at each ZT value as well as activity plots and their % change at each ZT. Since the l-LNvs have been shown to contribute to daytime activity in LD, a contribution to activity and therefore some inhibition of sleep is not too surprising - as mentioned above. So is this really specific for some "wake-maintenance zone?" And to what extent is this attributable specifically to this PDFR expression; two RNAis here are essential. And this is an issue throughout the paper, where single RNAs appear to be frequently sufficient to draw a conclusion. (Have at least two RNAis been used to validate the transcription factors identified near the end of the paper?)

3) tPDF and uas-PDFR could also be expressed in l-LNvs (specific gal4 is much better than c929) and in a pdf01 background. This would make the expression data more convincing. This would be much easier for the authors than a direct PDFR RNA assay from purified l-LNvs, which is another alternative. 

4) The numbers for live imaging are extremely low (5 neurons? That's maybe 2 brains). How many neurons react (1-4) and how reproducibly?

5) In Fig. 3B, why are the parental controls so low compared to CantonS in Fig. 3A? If statistics are done on all of the data in Fig. 3A and 3B together, would the PDFR mutant statistically differ from the parental controls of the RNAi experiment?

6) How often were the mating success experiments done? No error bars indicate n=1 (M&M also not clear). Also, the data are not very convincing, the WT controls in Fig. 3B are probably not significantly different from the han mutant in Fig 3A. How do the authors explain the index of 0? This suggests too many constructs rather than an effect of PDFR knockdown (pdf01 and han mutants are not sterile). The PDFR rescue should be done with lLNv-GAL4.

7) Labeling of images in Fig 5D is inverted.

8) How can a knockdown of dopamine receptors have no effect on the reactivity of the cell to dopamine? Once again a second RNAi is necessary to make this surprising claim.

9) I don't think the lifetime experiments (Fig. 4) are very compelling. I think of this more as manipulating activity than sleep. It makes sense that starving a more hyperactive fly is worse than a less active fly. Can this be done in parallel with another hyperactive strain?

10) The expression by different modalities is for sure interesting. The reader should learn if only this very special way to sleep deprive upregulates receptor expression or if a more traditional deprivation strategy can do the same thing.

11) Can we see the time of day data that underlies the decreased sleep (and presumably increased activity) in Fig. 5F, the sleep and activity profiles? Are the flies just generally more active, or is there some interesting time of day information? Perhaps the upregulation is just a stress-induced flight or fight response? A bit like enhancing foraging in response to starvation.

12) What's the sleep phenotype in the manipulations of Fig 6? Do nej knockdown flies phenocopy the absence of PDFR?

---

## [Decision Letter · Decision Letter 2]

9 Apr 2021

Dear Dr Shaw,

Thank you very much for submitting a revised version of your manuscript "Sleep-drive reconfigures wake-promoting clock circuitry to regulate adaptive behavior" for consideration as a Research Article at PLOS Biology. This revised version of your manuscript has been evaluated by the PLOS Biology editors, the Academic Editor and three of the original reviewers.

The reviews are appended below my signature. As you will see, reviewer 1 is satisfied with the revision, however reviewers 2 and 3 have a number of lingering concerns and note points requiring additional discussion and explanation. Most notably, both reviewers 2 and 3 have commented that it would be valuable to add data from pdf01 flies, even if this doesn’t phenocopy the pdfr mutants. Having discussed the reviewer comments with the Academic Editor, we think that it would be important for your revision to include this pdf01 data - whether they phenocopy Pdfr or not - and to discuss the implications of your findings. Given that this point has now been highlighted by all the reviewers, we think that this will be a question in the mind of many readers.

We are pleased to offer you the opportunity to address the remaining points from the reviewers in a revised version that we anticipate should not take you very long. We expect the revision in a month, however please do let us know if you need an extension to perform the additional pdf01 studies requested by the reviewers. We would be happy to accommodate such a request, as we think it is important that this data is added. We will then assess your revised manuscript and your response to the reviewers' comments and we may consult the reviewers again.

When addressing these last concerns, please also address the editorial requests, which I have included below my signature. 

Please email us (plosbiology@plos.org) if you have any questions or concerns, or to request an extension. At this stage, your manuscript remains formally under active consideration at our journal; please notify us by email if you do not intend to submit a revision so that we may end consideration of the manuscript at PLOS Biology.

**IMPORTANT - SUBMITTING YOUR REVISION**

*Resubmission Checklist*

*Published Peer Review*

*PLOS Data Policy*

*Blot and Gel Data Policy*

Sincerely,

Lucas Smith, Ph.D.,

Associate Editor,

lsmith@plos.org,

PLOS Biology

PLEASE ADDRESS THE REQUESTS BELOW:

DATA POLICY REQUEST:

Figure 1A-F; 2A-E; 3A-C,E-G; Fig 4A-D; Fig 5A-D,F-G; Fig 6A-F; Figure S1A-D; Fig S2B; Fig S4; Fig S5A-D

--Please also ensure that figure legends in your manuscript include information on where the underlying data can be found, and ensure your supplemental data file/s has a legend.

--Please ensure that your Data Statement in the submission system accurately describes where your data can be found.

EDITORIAL REQUESTS:

--I noticed that your materials and methods section has been included in a supplementary file. Please move this to the body of the main text.

-- Please take this last chance to review your reference list to ensure that it is complete and correct. If you have cited papers that have been retracted, please include the rationale for doing so in the manuscript text, or remove these references and replace them with relevant current references. Any changes to the reference list should be mentioned in the cover letter that accompanies your revised manuscript.

REVIEWS:

Reviewer #1: The authors have responded directly to the majority of the (all minor) concerns I expressed in my first review. I have no further concerns. It's an interesting, clear, and significant study.

Reviewer #2: Most of my concerns have been addressed. There is one lingering issue. I had realized that figure 2E showed sleep during WMZ. The value for controls in this panel are around 25-40, but on panel 2C CS shows a value of about 85. Those are rather big differences between different control flies. Where is this variability coming from? Is this a genetic background issue? Was pdfr-5304 backcrossed to CS? Controls for genetic backgrounds are good within figures, and a genetic rescue for the pdfr mutant phenotype is presented, so this does not seem to be a major concern, but some explanations are needed. 

Minor: The last paragraph of the results on p.9 contains a fragment of sentence that needs to be removed. 

I was also asked to look closely at the responses to the comments made by reviewer#4. Again, I think the authors did a good job responding to these comments. 

1) One issue that was brought up not just by reviewer #4 but also reviewer #1 is whether pdf0 phenocopies pdfr. This is a fair comment, and I agree with the authors' reply that pdf and pdfr mutants do not necessarily phenocopy each other, due to promiscuity of the pdfr receptor. However, this will be a question in the mind of many readers, and I would suggest that authors show pdf0 mutant data if they have it. 

2) I would like to comment on the issue of the need for two RNAis, which was brought up by reviewer #4 and #1. For pdfr phenotypes, I feel that a single RNAi phenocopying the pdfr mutant is sufficient, particularly since the authors can rescue this mutant. For the screen, when only one RNAi line is used, some words of caution would be worth adding. 

Reviewer #3: The authors have addressed my major comments--I have one remaining issue that I think would be beneficial for them to address. The authors have nicely clarified in their comments to the reviewers the definition of respecification. I suggest that they provide this level of clarity when they first introduce the concept in the paper and make clear the distinction with other examples in the literature.

With regards to reviewer 4's comment on pdf01, the authors make note of differences between pdf01 and pdfr mutants and their frustration with pdf01. I do think it would be valuable to include pdf01 data even if negative as it would highlight those differences and provide some insight into the relevant ligand for pdfr. Nonetheless the authors should make reference to the differences between pdf01 and pdfr mutants when discussing potential mechanisms by which pdfr might be functioning, i.e., pdfr may not be necessarily be working via PDF activation.

---

## [Editor Report · Decision Letter 3]

25 May 2021

Dear Dr Shaw,

Thank you for submitting your revised Research Article entitled "Sleep-drive reconfigures wake-promoting clock circuitry to regulate adaptive behavior" for publication in PLOS Biology. I have now obtained advice from the Academic Editor.

We think that the revised manuscript has, for the most part, addressed the concerns of the reviewers in the last round of review. However, we agree with the reviewers that it would be important to tone down statements regarding the RNAi screen on cell surface receptors, given that these are based on experiments with one RNAi line for each gene. Without using more than one RNAi line or transgene rescue, we think that you should not conclude positive effects nor negative effects from this screen, as there is the possibility that these effects were caused by off-target interactions or due to ineffective knockdown. The last paragraph of the results section should therefore be substantially revised, or even removed.

We will probably accept this manuscript for publication, provided you satisfactorily address the abovementioned point. IMPORTANT: Please also make sure to address the following data and other policy-related requests.

1) DATA REQUEST: Please provide, as a supplementary file or as a deposition in a publicly available repository, the data underlying each figure. Please be sure to reference this file in each figure legend and in your data availability statement - for example you might add to each figure legend the following statement "data underlying this figure can be found in S1_data. Please also include a legend for the supplementary file. I have included more information regarding this request below my signature.

2) Please move your materials and methods section into the main text of your manuscript (rather than including it as a supplementary file). 

3) In the interest of making your study accessible to as broad an audience as possible, we have been wondering if there might be a more colloquial term than "sleep-drive" that might be used in your title. However, we will leave it up to you if and how to change the title to address this point, as we would not want to disrupt its meaning. 

We expect to receive your revised manuscript within two weeks. 

*Published Peer Review History*

*Early Version*

Sincerely,

Lucas Smith, Ph.D.,

Associate Editor,

lsmith@plos.org,

PLOS Biology

PLEASE ADDRESS THE REQUESTS BELOW:

DATA POLICY REQUEST:

Figure 1A-F; 2A-E; 3A-C,E-G; Fig 4A-D; Fig 5A-D,F-G; Fig 6A-F; Figure S1A-D; Fig S2B; Fig S4; Fig S5A-D

**IMPORTANT: 

--Please also ensure that figure legends in your manuscript include information on where the underlying data can be found, and ensure your supplemental data file/s has a legend.

--Please ensure that your Data Statement in the submission system accurately describes where your data can be found.

---

## [Editor Report · Decision Letter 4]

15 Jun 2021

Dear Dr Shaw,

On behalf of my colleagues and the Academic Editor, Bing Ye, I am pleased to say that we can in principle offer to publish your Research Article "Sleep-drive reconfigures wake-promoting clock circuitry to regulate adaptive behavior" in PLOS Biology, provided you address any remaining formatting and reporting issues. These will be detailed in an email that will follow this letter and that you will usually receive within 2-3 business days, during which time no action is required from you. Please note that we will not be able to formally accept your manuscript and schedule it for publication until you have made the required changes.

When making these final changes, we also ask that you provide legends for your new Supplementary data files. 

Please also take a minute to log into Editorial Manager at http://www.editorialmanager.com/pbiology/, click the "Update My Information" link at the top of the page, and update your user information to ensure an efficient production process.

PRESS

Sincerely, 

Lucas Smith, Ph.D. 

Senior Editor 

PLOS Biology

lsmith@plos.org